# Active learning-assisted directed evolution

Jason Yang [1,6], Ravi G. Lal[1,6], James C. Bowden[2,5], Raul Astudillo[2], Mikhail A. Hameedi [3], Sukhvinder Kaur[4], Matthew Hill[4], Yisong Yue [2] ✉ & Frances H. Arnold [1,3] ✉

Directed evolution (DE) is a powerful tool to optimize protein fitness for a specific application. However, DE can be inefficient when mutations exhibit non-additive, or epistatic, behavior. Here, we present Active Learning-assisted Directed Evolution (ALDE), an iterative machine learning-assisted DE workflow that leverages uncertainty quantification to explore the search space of proteins more efficiently than current DE methods. We apply ALDE to an engineering landscape that is challenging for DE: optimization of five epistatic residues in the active site of an enzyme. In three rounds of wet-lab experimentation, we improve the yield of a desired product of a non-native cyclopropanation reaction from 12% to 93%. We also perform computational simulations on existing protein sequence-fitness datasets to support our argument that ALDE can be more effective than DE. Overall, ALDE is a practical and broadly applicable strategy to unlock improved protein engineering outcomes.

Protein engineering is an optimization problem, where the goal is to find the amino acid sequence that maximizes "fitness," a quantitative measurement of the efficacy or functionality for a desired application, from chemical synthesis to bioremediation and therapeutics[1]. Protein fitness optimization can be thought of as navigating a protein fitness landscape, a mapping of amino acid sequences to fitness values, to find higher-fitness variants[2]. However, since protein sequence space is vast, as a protein of length $N$ can take on $20^N$ distinct sequences and functional proteins are vanishingly rare, finding an optimal sequence is hard. Because functional proteins are surrounded by other functional proteins one mutation away[3], protein engineers often use directed evolution (DE) to optimize protein fitness[4,5].

In its simplest form, DE involves accumulating beneficial mutations by searching through sequences near one that exhibits some level of desired function for variants that exhibit enhanced performance on a target fitness metric (Fig. 1A). This approach can be thought of as greedy hill climbing optimization across the protein fitness landscape (Fig. 1B). DE is limited because screening for performance can only explore a small, local region of sequence space. Additionally, taking one mutational step at a time can cause the experiment to become stuck at a local optimum, especially on rugged

protein fitness landscapes where mutation effects exhibit epistasis[6]. Machine learning (ML) techniques offer a pathway to circumvent these obstacles, providing strategies to more efficiently navigate these complex landscapes[7–11].

While supervised ML has been used to propose ideal combinations of mutations–such as in ML-assisted DE (MLDE)[12,13]–these approaches are often limited to small design spaces as they do not take advantage of the fundamentally iterative manner in which protein engineering can take place in real-world applications. By contrast, active learning is an ML paradigm that gathers data iteratively using a supervised model which is, in turn, updated as new data are acquired (Fig. 1C). By leveraging uncertainty quantification to choose which variants should be tested at each step, active learning has the potential to unlock improved engineering outcomes (Fig. 1D)[14–18]. Approaches related to active learning have been used in the wet lab to optimize artificial metalloenzymes, nucleases, and other proteins[19–23]. Past work has also explored the use of Bayesian optimization (BO), a particular class of active learning algorithms, to experimentally improve the thermostability of protein chimeras[24,25] and to optimize proteins with one to several mutations[14,26,27]. However, few studies have explored the utility of active learning methods in comparison to DE, especially

[1]Division of Chemistry and Chemical Engineering, California Institute of Technology, Pasadena, CA, USA. [2]Division of Engineering and Applied Sciences, California Institute of Technology, Pasadena, CA, USA. [3]Division of Biology and Biological Engineering, California Institute of Technology, Pasadena, CA, USA. [4]Elegen Corp, 1300 Industrial Road #16, San Carlos, CA, USA. [5]Present address: Computer Science, University of California-Berkeley, Berkeley, CA, USA. [6]These authors contributed equally: Jason Yang, Ravi G. Lal. ✉e-mail: yyue@caltech.edu; frances@cheme.caltech.edu

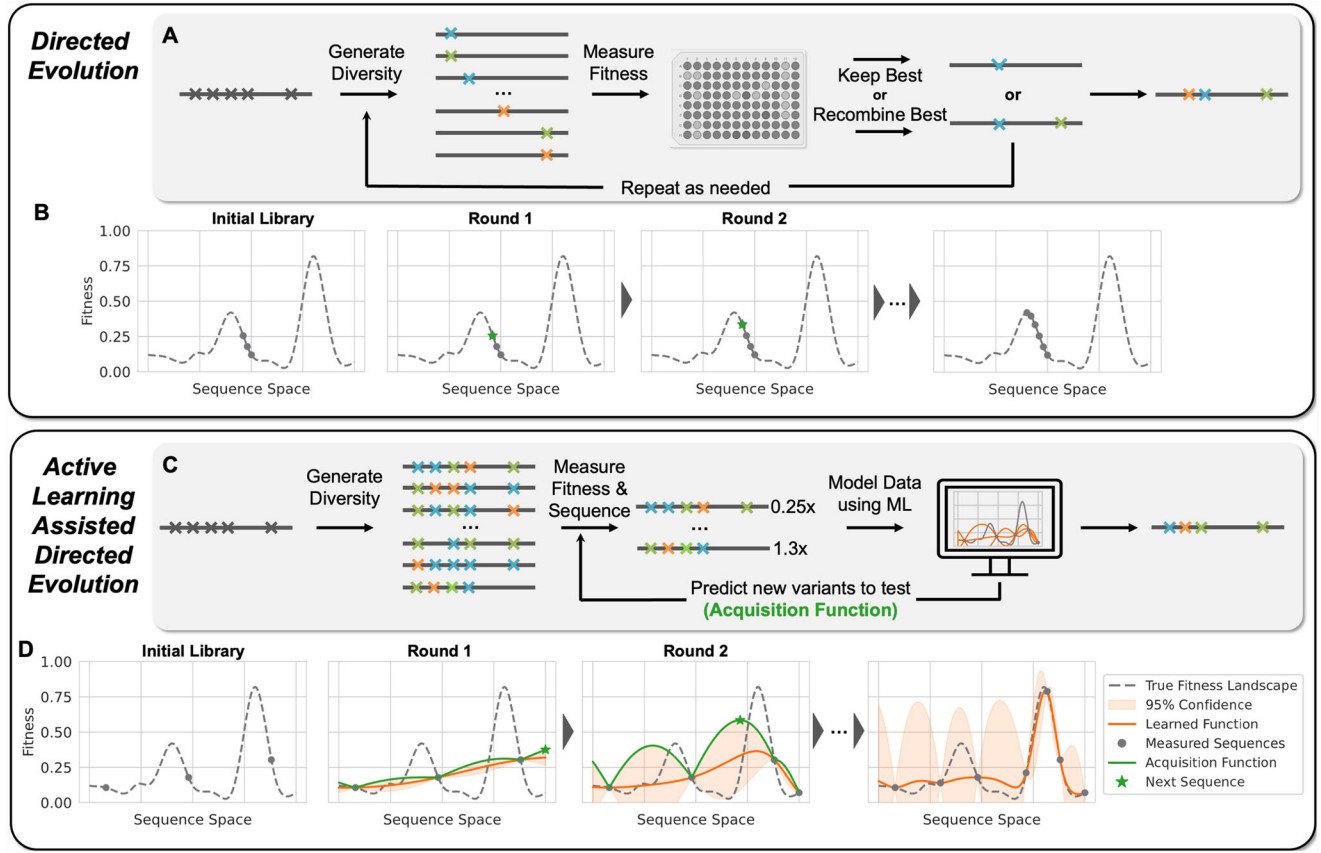

**Fig. 1 | Conceptual differences between DE and ALDE. A** A common workflow for DE, where a starting protein is mutated and fitnesses of variants are measured (screened). The best variant is used as the starting point for the next round of mutation and screening, until desired fitness is achieved. **B** Conceptualization of DE as greedy hill climbing optimization on a hypothetical protein fitness landscape. **C** Workflow for ALDE. An initial training library is generated, where $k$ residues are mutated simultaneously (for example $k = 5$). A small subset of this library is randomly picked, after which the variants are sequenced and their fitnesses are screened. A supervised ML model with uncertainty quantification is trained to learn a mapping from sequence to fitness. An acquisition function is used to propose new variants to test, balancing exploration (high uncertainty) and exploitation (high predicted fitness). The process is repeated until desired fitness is achieved. **D** Conceptualization of active learning on a hypothetical protein fitness landscape. Active learning is often more effective than DE for finding optimal combinations of mutations. In these conceptualizations, a single sequence is queried in each round, but in practical settings, active learning operates in batch where multiple sequences are tested in each round.

where epistatic effects are prevalent[20,28]. In addition, understanding of the practical role of uncertainty quantification in the context of deep learning[29–31] and high-dimensional[32] representations learned from protein language models[33,34] is limited.

  To address the limitations of existing methods, we introduce Active Learning-Assisted Directed Evolution (ALDE), a computationally assisted workflow for protein engineering that employs batch Bayesian optimization. ALDE alternates between collecting sequence-fitness data using a wet-lab assay and training an ML model to prioritize new sequences to screen in the wet lab (Fig. 1C); it resembles existing wet-lab mutagenesis and screening workflows for DE and is generally applicable to any protein engineering objective. In this study, we use ALDE to find the ideal combination of five mutations in the active site of a biocatalyst based on a protoglobin from *Pyrobaculum arsenaticum* (*Par*Pgb) for performing a non-native cyclopropanation reaction with high yield and stereoselectivity. We chose this model system because the residues of interest are in close structural proximity and there is evidence of negative epistasis, which hinders DE. After performing three rounds of ALDE (exploring only ~0.01% of the design space), the optimal variant has 99% total yield and 14:1 selectivity for the desired diastereomer of the cyclopropane product. The mutations present in the final variant are not expected from the initial screen of single mutations at these positions, demonstrating that the consideration of epistasis through ML-based modeling is important. We solidify our

argument that ALDE is more effective than DE by computationally simulating ALDE on two combinatorially complete protein fitness landscapes. We also provide an extensive analysis of the effects of protein sequence encodings, models, acquisition functions, and uncertainty quantification for protein fitness optimization, to determine best practices for real-world engineering campaigns. In short, we find that frequentist uncertainty quantification works more consistently than typical Bayesian approaches, and incorporating deep learning does not always boost performance. Ultimately, we demonstrate that ALDE is a practical and effective tool for navigating protein fitness landscapes and provide experimental and computational tools (https://github.com/jsunn-y/ALDE) so that the method is easy to use and broadly applicable.

## Results

### Practical implementation of ALDE

Broadly, ALDE alternates between library synthesis/screening in the wet lab to collect sequence-fitness labels and computationally training an ML model to learn a mapping from sequence to fitness in order to suggest a new batch of sequences to test (Fig. 1C), resembling batch BO. Before beginning ALDE, a combinatorial design space on $k$ residues is defined, corresponding to $20^k$ possible variants. The choice of $k$ will vary depending on the system, as larger values of $k$ can consider a greater extent of epistatic effects (allowing for better possible

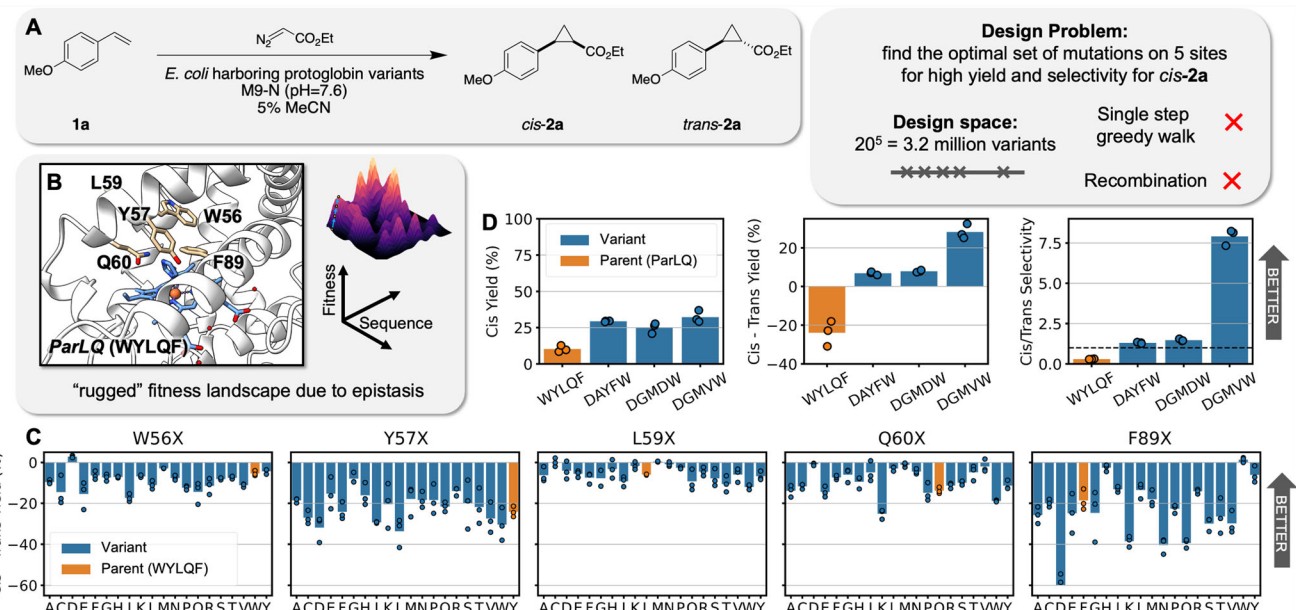

**Fig. 2 | A challenging, epistatic protein design space: optimization of five active site residues in *Par*Pgb. A** Our objective was to optimize an enzyme to catalyze the formation of the *cis* product of a cyclopropanatiom reaction with high yield and high selectivity, which we quantify in a single value as *cis* − *trans* Yield. **B** The parent protein ParLQ is two mutations (W59L and V60Q) away from the wild-type ParPgb sequence. Five residues in the active site of ParLQ which were likely to exhibit epistasis were targeted: W56, Y57, L59, Q60, and F89. **C** The single mutations from parent at the five targeted sites do not offer significant improvements to the objective of *cis* − *trans* Yield. Very few single-mutation variants have the desired selectivity (positive *cis* − *trans* Yield), and it would not be obvious which variant to

take forward in a DE campaign. Parent yields vary between runs but consistently show moderate yield and selectivity for the *trans* product. **D** Various recombinations of ideal single mutations are not effective proteins for the desired objective (*cis*–*trans* Yield), and related metrics such as *cis* Yield and *cis/trans* Selectivity. DAYFW, DGMDW, and DGMVW are the ideal combinations of single mutations naively predicted to have the highest *cis* Yield, *cis* − *trans* Yield (objective), and *cis/trans* Selectivity, respectively. Yields were measured in biological triplicate. Overall, these results suggest an optimization problem that is challenging for standard DE methods. Source data are provided as a Source Data file.

outcomes) but will likely require collecting more data to find an optimal variant. First, those *k* residues are simultaneously mutated, and an initial round of sequence-fitness data is collected in the wet lab. ALDE is compatible with low-N, batch protein engineering settings where tens to hundreds of sequences are screened in each round. The collected sequence-fitness data are then used to computationally train a supervised ML model that can predict sequence from fitness. Different ways to encode protein sequence numerically and different types of models which can provide uncertainty quantification are analyzed in this study. Afterward, an acquisition function is applied to the trained model to rank all sequences in the design space, from most to least likely to have high fitness. Several acquisition functions are evaluated in this study, to balance *exploration* of new areas of protein space with *exploitation* of variants that are predicted to have high fitness (Fig. 1D). The computational component of ALDE can be performed using the codebase at https://github.com/jsunn-y/ALDE. For the next round of ALDE, the top N variants from the ranking are then assayed in the wet-lab to provide additional sequence-fitness data, and the cycle is repeated until fitness is sufficiently optimized.

## The active site of ParPgb is a challenging design space for standard DE

To initiate wet lab studies with ALDE, we identified a target enzymatic activity on a protein design space that would be difficult to engineer with simple DE methods. Enzyme-catalyzed carbene transfer reactions have the potential to be useful in many synthetic chemistry applications, and thus we decided to focus on the cyclopropanation of 4-vinylanisole (**1a**) using ethyl diazoacetate (**EDA**) as a carbene precursor to afford the 1,2-disubstituted cyclopropanes *trans*-**2a** and *cis*-**2a** (Fig. 2A). Enzyme engineering for styrenyl cyclopropanation poses a stimulating challenge for evolution toward two properties, higher

yield *and* improved selectivity toward one of the diastereomers of the cyclopropane product. While this non-native chemistry has been demonstrated with cytochromes P411[35], we decided to engineer this activity in a protoglobin. Protoglobins are archaeal hemoproteins, which are attractive engineering targets due to their high thermostability (T$_{50}$ ~ 60 °C), small size (~200 amino acids)[36], and ability to perform novel carbene and nitrene transfer chemistries[37–40]. After screening a diverse set of protoglobins, including wild-types and engineered homologs, for cyclopropanation activity (Fig. S31 of Supplementary Information), we decided to proceed with *Par*Pgb W59L Y60Q (ParLQ) as a starting point (parent) for ALDE. Because our goal was to arrive at a variant with high yield and high selectivity for the *cis*-product, we defined the objective to be explicitly optimized as the difference between the yield of *cis*-**2a** and the yield of *trans*-**2a**. The ParLQ variant demonstrates only moderate cyclopropanation yield (~40% yield) and stereoselectivity (3:1 preferring *trans*-**2a**) under screening conditions, and no known protein variant of *Par*Pgb has high fitness for our objective.

Based on previous engineering studies using protoglobin scaffolds, we selected five active-site residues (W56, Y57, L59, Q60, and F89; WYLQF) positioned above the distal face of the heme cofactor, which display epistatic effects and are known to impact non-native activity (Fig. 2B)[38,39]. Single-site saturation mutagenesis (SSM) was performed at these sites, and variants were screened by gas chromatography for their cyclopropanation products. None of the screened mutants demonstrated a significant, desirable shift in the value of the objective (Fig. 2C) or related metrics such as *cis* yield and *cis/trans* selectivity (Figs. S32–S46). Given these data, a protein engineer might opt to perform a simple recombination of all positive variants to exploit the typically additive character of mutations[41]. However, in our recombination studies of the single-site mutants with the highest fold-

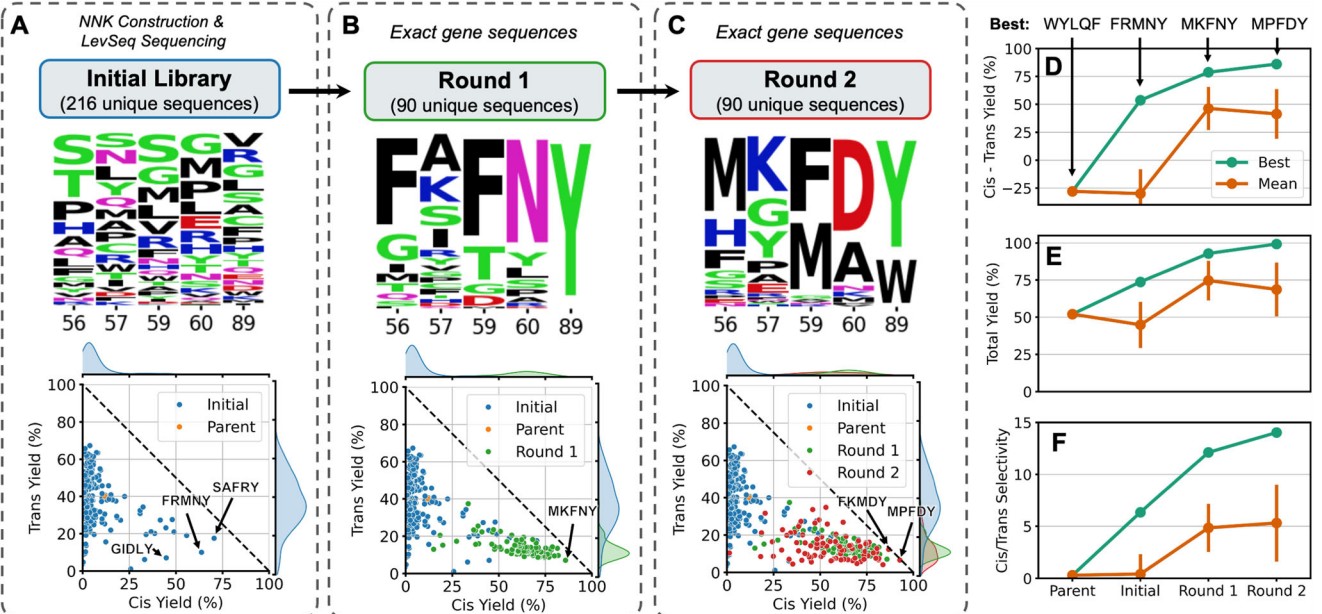

**Fig. 3 | ALDE optimization trajectory on the *Par*Pgb active site.** The optimization campaign started with (**A**) constructing an initial library with mutations at all five sites under study using NNK degenerate codons, randomly selecting 384 for screening for product formation, and mapping to sequences using LevSeq. This was followed by two rounds of ALDE—(**B**) Round 1 and (**C**) Round 2. In Round 1 and Round2, exact genes were ordered as ENFINIA DNA produced by Elegen Corp. and screened for product formation. For each round, we present the distribution of amino acids sampled at each site and the distribution of yields for the *cis* and *trans* products, with a few of the top-performing variants labeled. Overall improvement in (**D**) *cis* – *trans* Yield, (**E**) Total Yield, and (**F**) *cis/trans* Selectivity over several rounds of ALDE for the best variant in each round and the mean across variants in each round. The best variant in each round, defined by the objective of *cis* – *trans* Yield is labeled. Error bars indicate the standard deviation across variants in the round. Yields were measured in biological triplicate. Source data are provided as a Source Data file.

change in *cis* yield (DAYFW), the objective (DGMDW), or the selectivity (DGMVW), respectively, we did not observe a variant which generated *cis*-**2a** with high yield and selectivity (Fig. 2D). Overall, these findings suggest that our design problem is quite challenging for standard DE approaches.

## Using ALDE to efficiently optimize ParPgb for a non-native carbene transfer reaction

With the design space confined to five residues and a well-defined objective, we began an ALDE engineering campaign. First, we synthesized an initial library of ParLQ variants which were mutated at all five positions under study (Fig. 3A). Mutants in this library were generated through sequential rounds of PCR-based mutagenesis methods utilizing NNK degenerate codons. We elected to use random selection from this library because we did not know if any zero-shot predictors might enrich the starting library with useful variants[8,13]. In fact, retrospective analysis of the initial library revealed that our objective is not strongly correlated with conventional zero-shot predictors[13,42], likely because the objective involves non-native chemistry, for which fitness is not sufficiently captured by evolutionary or stability-based metrics alone (Fig. S72 of Supplementary Information). Four 96-well plates of these random variants were picked and sequenced using the LevSeq long-read pooled sequencing method (Figs. S7–10 of Supplementary Information)[43], yielding 216 unique variants without stop codons. Screening revealed that nearly all of the variants had higher cyclopropanation activity than free-heme background activity, likely because *ParLQ* was moderately active to begin with, and its high thermostability allows it to tolerate multiple mutations. The majority of variants displaying improved cyclopropanation yield strongly favored formation of *trans*-**2a**; however, several of the randomly selected sequences were capable of forming *cis*-**2a** in much higher yield than any previously tested ParLQ variant (Fig. 3B). Notably, the F89Y mutation was particularly important for inverting selectivity to

favor the *cis*-**2a**, but only in the context of certain mutations at positions 56, 57, 59, and 60.

The ALDE computational package was used to train a predictive model on sequences and labels in the initial 216-member library and to suggest sequences for testing based on our acquisition function. Based on our extensive computational simulations (described in the following section), we decided to use the DNN ensemble with one-hot encoding of the five targeted residues for model training and Thompson sampling as the acquisition function. Genes encoding the top 90 amino acid sequences, optimized for expression in *E. coli*, were prepared by exact DNA synthesis for screening (Round 1, Fig. 3B). Details regarding DNA sequence design are described in the included supplementary materials. Subsequent activity screening of these sequences in triplicate showed that nearly a third of Round 1 sequences met the objective better than the best variant in the initial, randomly selected set (Fig. 3B). The best variant in the Round 1 library, MKFNY (W56M Y57K L59F Q60N F89Y), demonstrated a total cyclopropanation yield of 93% and a *cis:trans* selectivity ratio of 12:1.

We then gave the newly collected data back to the ALDE computational algorithm for a second round of active learning. The top 90 predicted sequences were again synthesized and tested exactly as before (Round 2, Fig. 3C). Interestingly, the model *explored* the sequence space more in this round, as reflected in the expanded mutational diversity present in Round 2 and the increased variance in the activities of these sequences (both reaction yield and diastereoselectivity) (Fig. 3D–F). Impressively, the top-performing variant among these sequences (MPFDY) displayed a total cyclopropane yield of 99% and a 14:1 *cis:trans* selectivity ratio. None of the mutations in MPFDY obviously optimized the objective in the single-site mutagenesis studies (Fig. 2C); they work together, however, to deliver an optimal variant. Furthermore, after screening the reaction products of all predicted variants with chiral gas chromatography methods, we found

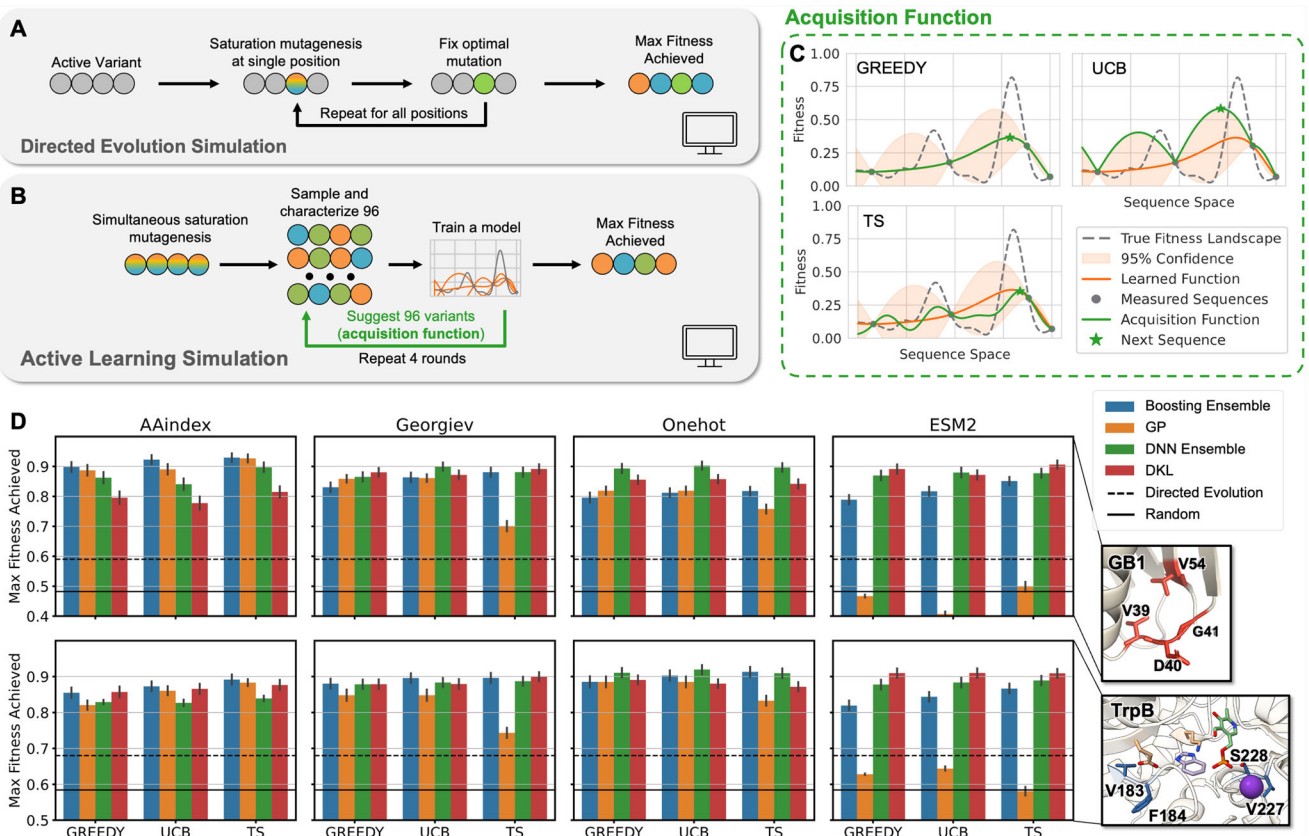

**Fig. 4 | Performance of simulated ALDE campaigns on two combinatorially complete protein datasets, GB1 and TrpB. A** Each DE simulation as a greedy single-step walk on four residues, where each residue is fixed to the optimal mutation until all four residues have been iterated across. DE simulations start from every variant that has some measurable function, with all 24 possible orderings of four residues simulated. **B** Each ALDE simulation starts from a random sample of 96 variants on the 4-site landscape, with four rounds of learning and proposing new sequences to test, each with 96 protein variants. **C** Hypothetical visualization of the three acquisition functions explored in this work: greedy, upper confidence bound (UCB), and Thompson sampling (TS). **D** ALDE for four encodings, four models, and three acquisition functions generally outperforms the average DE simulation and random sampling on the GB1 and TrpB datasets. Performance is quantified as the normalized maximum fitness achieved by the end of the ALDE campaign. Error bars indicate standard deviation across 70 random initializations. Source data are provided as a Source Data file.

that all of these sequences were generally capable of generating *cis*-**2a** in high enantiopurity (Fig. S71 of Supplementary Information).

Having concluded the ALDE-based evolutionary campaign with substrate **1a**, we sought to understand the substrate scope of the sequences explored in this project. We screened eight styrene derivatives (**1b–1i**) for cyclopropanation using the sequences from Round 2 of ALDE (Fig. S50 of Supplementary Information). The variants show different yields for each of the substrates, even though some of these substrates differed from **1a** only by a single atom. Nevertheless, for every substrate, nearly all of the Round 2 variants were higher yielding and more selective for their respective *cis*- diastereomers than the parent protein, ParLQ (Figs. S52–67 of Supplementary Information). Interestingly, the top-performing variants for each substrate differed in sequence from MPFDY, the top enzyme for **1a** cyclopropanation. For all the predicted variants in Round 1 and 2 of ALDE, sequences were confirmed with LevSeq, and the yield and selectivity of the top variant from each round was validated in vial format, showing good overall consistency (Fig. S30b).

**Computational simulations on combinatorial protein datasets support the utility of ALDE**
The design choices used for the wet-lab ALDE campaign were determined by performing computational simulations on two combinatorial landscape datasets for GB1[44] and TrpB[45]. On these landscapes, fitnesses have been measured experimentally for nearly all of the $20^4 = 160,000$

variants in a library where four amino acid residues were mutated to all possible amino acids. GB1 refers to the B1 domain of protein G, an immunoglobulin binding protein where fitness is measured by binding affinity-based sequence enrichment. The fitness of TrpB, the β-subunit of tryptophan synthase, was measured by coupling growth to the rate of tryptophan formation. Our baseline was DE greedy walk, where one residue was mutated to all possible amino acids, the best mutation was fixed, and the process was repeated at each of the residues under study (Fig. 4A). DE simulations were performed from all active variants as starting points, using all 24 possible orders to enumerate the residues under interest.

The ALDE simulation consisted of batch BO, as shown in Fig. 4B. In each simulation, a random batch of 96 initial samples was selected, followed by four rounds of 96 samples each, with the surrogate model retrained and proposing new samples (via the acquisition function) in each round. This simulation setup was chosen to closely imitate a real wet-lab active learning campaign. The different parameters explored for ALDE, including encodings, models, and acquisition functions, are summarized in Table 1. We expanded the analysis beyond Gaussian process (GP) models, which are the typical surrogate models for BO, to deep kernel learning (DKL) models[29,31] and frequentist models based on boosting and deep neural network (DNN) ensembles. This was motivated by the rise of high dimensional encodings of protein sequences, such as those from protein language models (i.e. ESM2[33]), which have shown utility in certain property prediction tasks[46,47].

**Table 1 | Summary of encodings of protein sequences, models, and acquisition functions tested in this work**

| Encoding | Dimension per Residue | | Description |
|---|---|---|---|
| AAIndex | 4 | | Continuous fixed amino acid descriptors |
| Georgiev[71] | 19 | | Continuous fixed amino acid descriptors |
| Onehot | 20 | | Categorical (which amino acid) |
| ESM2[33] | 1280 | | Learned embedding from a protein language model (ESM2 with 650 million parameters) |
| **Model** | Bayesian? | Deep Learning? | Description |
| Boosting Ensemble | N | N | An ensemble of 5 boosting models |
| Gaussian Process (GP) | Y | N | A collection of continuous functions described by a posterior |
| DNN Ensemble | N | Y | An ensemble of 5 multilayer perceptrons (deep neural networks, DNNs) |
| Deep Kernel Learning (DKL)[29] | Y | Y | A GP on the last layer of a deep neural network |
| **Acquisition Function** | Deterministic? | | Description |
| Greedy | Y | | Acquires the maximum value of the mean from the posterior |
| Upper Confidence Bound (UCB) | Y | | Acquires the maximum value of a certain confidence interval from the posterior (tuned by a hyperparameter) |
| Thompson Sampling (TS) | N | | Acquires the maximum value of a random function sampled from the posterior |

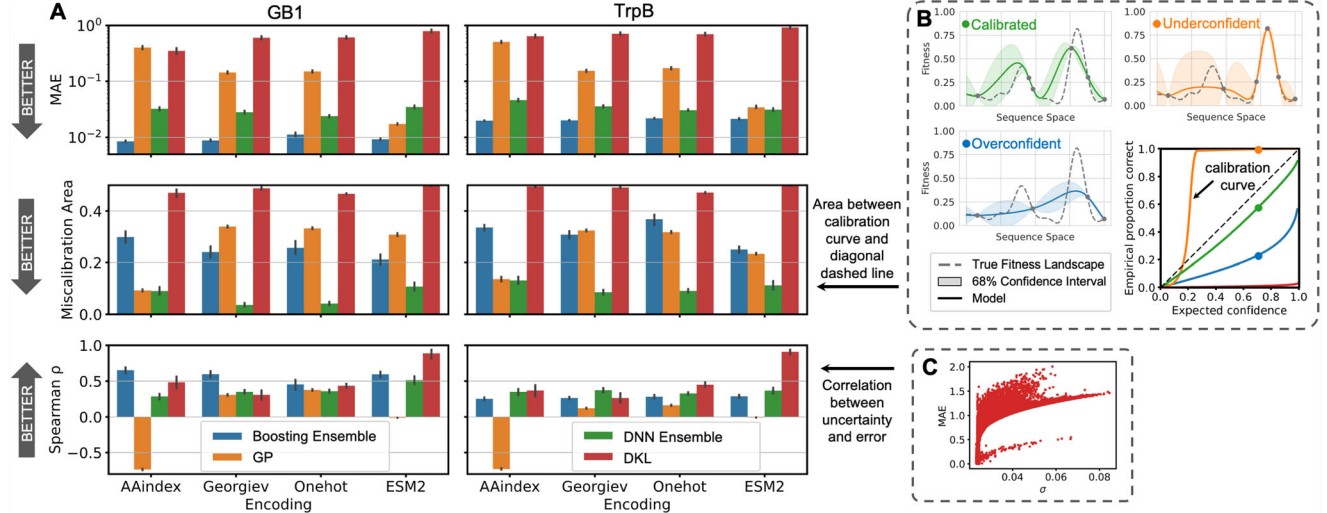

**Fig. 5 | Analysis of uncertainty quantification on simulated ALDE campaigns. A** Metrics used to evaluate how well calibrated each of the four models are for four encodings. Metrics for evaluation are the mean absolute error (MAE), the miscalibration area for the calibration curve, and the Spearman correlation between uncertainty and error. All metrics are calculated based on all measured points in the combinatorial design space. All results are based on the campaigns using UCB as the acquisition function, during the final round of the campaign. Error bars indicate standard deviation across 70 random initializations. **B** Visualizations of three hypothetical models with underconfident, calibrated, and overconfident uncertainties, and their respective calibration curves. **C** Visualization of how the Spearman correlation between uncertainty and error is calculated. Source data are provided as a Source Data file.

Visualizations of the acquisition functions (greedy, upper confidence bound (UCB), and Thompson sampling (TS)) on hypothetical models are given in Fig. 4C, with more details in **Methods**.

The performance of each simulated ALDE campaign was quantified as the maximum fitness achieved at the end of the campaign, normalized to the variant with maximum fitness in the design space (Fig. 4D). Full optimization trajectories at each iteration of the campaign are provided in Fig. S73. We conclude that active learning can significantly outperform the average performance of DE and random sampling, and results are consistent across the two different protein datasets. ALDE is competitive with similar methods[14,15] and also outperforms a single round of MLDE (Fig. S74 of Supplementary Information). Higher dimensional encodings (Onehot and ESM2) generally work better with deep learning-based models (DNN Ensemble and DKL), while non-deep learning models might learn better from low dimensional AAIndex and Georgiev parameters. The simulations further suggest that encodings from protein language models may not offer much benefit, which corroborates previous findings[13] but stands in contrast to other protein properties that can be predicted more

effectively by transfer learning from protein language models[21,46,47]. We find that ESM2 encodings cannot be used by GPs, likely because they are too high dimensional. Other studies suggest that using the right length-scale priors with GP can enable them to work in these settings[48,49]; while we did not observe this same effect for our application, further exploration may be interesting here. In our acquisition functions, samples in the batch were sampled independently of each other. We also explored batch expected improvement[50], but this ran extremely slowly without noticeable improvement in performance. Overall, the frequentist ensemble models perform the most consistently across different encodings.

To better understand which models are the most advantageous, we assessed how well calibrated their uncertainties were (Fig. 5A). For a calibrated model, an n% confidence interval should contain n% of true labels across different values of n, which can be evaluated and visualized based on a calibration curve. Hypothetical calibrated, underconfident, and overconfident models are visualized in Fig. 5B, with their associated calibration curves. The calibration curves for different encodings and models are given in Fig. S75. The area between a

calibration curve and perfect calibration (dashed line) is defined as its miscalibration area, which should be low. Another way to measure uncertainty calibration is by measuring the Spearman correlation between uncertainty from the model (σ) and the mean absolute error from the model (MAE), which should be high.

Overall, the Boosting and DNN Ensembles have the lowest MAEs, which suggests that they are the most accurate models (Fig. 5A). DNN ensembles have the lowest miscalibration areas, suggesting that they are the most calibrated and best models overall. These results are generally consistent across encodings and datasets, with a few outliers. In general, calibrated uncertainty is desirable[51,52], and it is thought that it is important to understand how calibration shifts when extrapolating beyond the training set[53,54]. However, in this study, we find that performance in ALDE simulations (by max fitness achieved) is not necessarily correlated to how calibrated the uncertainties are for each model. For example, DKL performs the best for the ESM2 encoding, but these models have the least calibrated uncertainties and the highest MAEs. The poor calibration of DKL models may result from some mode collapse where out-of-distribution inputs are mapped close to the training representations by the neural network[55]. Because calibration is measured on the entire combinatorial design space, it may not directly correspond to the ability to find an optimal variant.

## Discussion

Overall, ALDE is an effective method for navigating protein fitness landscapes, and it offers several advantages compared to DE. First, ALDE can unlock engineering outcomes not possible with simple DE. By considering multiple interacting positions, ALDE can search for combinations of mutations which may demonstrate desirable epistatic effects[56,57] and reduce the risk of getting trapped at a local optimum. We demonstrated the advantage of ALDE on *Par*Pgb as a particular wet-lab case study—though proof is not possible without testing every DE greedy single-step walk (which is experimentally intractable in the wet lab). Computational simulations of ALDE support this conclusion, as ALDE consistently outperforms MLDE and DE baselines. While ALDE and MLDE[12,13] are similar conceptually and practically, MLDE only uses greedy acquisition, whereas ALDE can consider model uncertainty, which is potentially useful for exploring larger design spaces. Compared to previous computational studies on BO for protein variants[14,26], our study examines a more comprehensive range of encodings, models, and acquisition functions, and it introduces analysis of the role of calibrated uncertainty quantification. Interestingly, we found that frequentist ensembles work the best in terms of performance and uncertainty quantification[30,58], rather than Bayesian approaches such as typical GP models used in BO. Other ways to quantify uncertainty and improve overall performance could be explored in the future[16,30]. Overall, classical notions of uncertainty quantification seem to play a more limited role than expected in these real-world applications. In a related study, we show that ALDE can be combined with various zero-shot predictors and that our findings here still hold for 16 different protein-fitness landscapes, including those with fewer active variants[59].

In the wet-lab engineering campaign, we were pleased to find that ALDE enabled access to a compilation of enzymes which, when considered together, demonstrate a broader substrate scope than that of a single enzyme. By contrast, DE is limited because it often "locks" one into high yield for only a single substrate or closely related ones, as the final variant is generally a single optimized sequence. Here, we observe an emergent advantage inherent to ALDE: since sequences that balance *exploration* and *exploitation* for a given task are proposed, they can be serendipitously proficient at related tasks.

ALDE is enabled by several recent advancements in biotechnology. For the initial library constructed using degenerate codons, high-throughput sequencing was necessary to identify the sequences of variants in each well. For this work we utilized LevSeq[43,60], a method that leverages real-time nanopore sequencing. Furthermore, rapid and reliable access to directly synthesized DNA was instrumental to the speed with which evolution was performed. The ALDE workflow was significantly enhanced with (1) the delivery of exact genes in one week, which shortened time between rounds of evolution, (2) the high fidelity of the delivered gene products meant that no sequencing was required for Rounds 1 and 2 of ALDE, and (3) no over-screening was needed because the exact sequences were arrayed individually. Overall, the time and screening cost of the wet-lab engineering campaign with ALDE was lower than for a greedy walk strategy with DE. A total of six 96-well plates were screened before arriving at a final variant: four plates of random variants, and two plates of predicted sequences within three rounds. By comparison, a greedy walk with DE would have required around five rounds of evolution–typically one mutation is accumulated in each round of a DE greedy walk–with increased screening in the later rounds, which would require greater experimental resources such as chemical reagents and analysis time. We expect that exact gene synthesis will be increasingly important for powering active learning workflows in protein engineering.

In this work, we illustrated ALDE's power for simultaneously increasing the activity and selectivity of an enzyme for a non-natural reaction, but ALDE is a general workflow that can be used for a broad range of protein engineering applications. Additionally, ALDE could be integrated into robotic systems for automated and efficient protein engineering workflows, and library design could utilize tools such as DeCOIL[61]. While we only engineered on five residues in this study, ALDE should naturally extend to even larger design spaces on more residues, as long as assay-labeled data is collected on variants with mutations spread across those residues. Determining the residues to target is an open challenge: these residues should tolerate mutations and have the potential to increase the fitness of interest. Initial domain knowledge, evolutionary conservation, or initial mutational screening may be useful here. Library design could also benefit from limiting the number of simultaneous mutations or using zero-shot scores[13,19]. Despite our in silico simulations using combinatorially complete datasets and wet-lab demonstration of ALDE, it remains an open question how generally our findings can be applied to the engineering of other enzymatic systems. Further work is needed to understand how the number of samples and/or rounds required to achieve successful engineering outcomes will increase (linearly or exponentially) with the number of sites explored simultaneously and how epistasis affects this. Future work here may also involve generative modeling if it is not possible to enumerate the acquisition function on the entire design space. Overall, accompanied by a user-friendly codebase, ALDE is a broadly applicable tool that can unlock more efficient and effective protein engineering.

## Methods

### Cloning of random ParPgb variants

**Cloning for single site-saturation mutagenesis.** Chemically competent *Escherichia coli* (*E. coli*) cells (T7 Express Competent *E. coli*) were purchased from New England Biolabs (NEB, Ipswich, MA). Phusion polymerase and *Dpn*I were purchased from NEB. SSM experiments were performed using primers bearing degenerate codons (NNK) using a modified QuikChange™ protocol[62]. The PCR conditions were (final concentrations): Phusion HF Buffer 1x, 0.2 mM dNTPs each, 0.5 μM of forward primers, 0.5 μM reverse primer, and 0.02 U/μL of Phusion polymerase. The standard Phusion PCR protocol was used. Upon completion of PCRs, the remaining template was digested with *Dpn*I. Gel purification was performed with a Zymoclean Gel DNA Recovery Kit (Zymo Research Corp, Irvine, CA). The purified PCR product was then assembled using the Gibson assembly protocol[63].

**Transformation of single site mutants.** 96-well deep-well plates are shaken in an INFORS HT Multitron Shaker in all instances. The assembly products obtained were used to transform T7 Express

Competent *E. coli* (High Efficency) cells (NEB, Ipswich, MA) following the recommended protocol. Upon heat-shock transformation, mixtures were recovered in 0.4 mL Luria-Bertani medium (LB) (Research Products Int.), after which the cells were incubated at 37 °C with shaking at 220 rpm for 30 min before being plated on LB-agar plates with 100 µg/mL ampicillin (LB-amp agar plates). Single colonies from LB-agar plates were picked using sterilized toothpicks, which were used to individually inoculate 400 µL of LB containing 100 µg/mL of ampicillin (LB-amp) in 2 mL 96-well deep-well plates. The plates were incubated at 37 °C and shaken at 220 rpm for 16-18 h. The following morning 50 µL of preculture from each well were added to the wells of a 96-well flat-bottom tissue culture plate (ThermoFisher) preloaded with 50 µL of 50% glycerol solution. These glycerol stocks were stored at −80 °C for future inoculation. Additionally, the sequences of protoglobin genes contained in every well were sequenced using the evSeq protocol[60].

**Cloning for multisite-saturation mutagenesis.** Mutations were simultaneously incorporated as with single SSM using the ParLQ_quadNNK primers (Table S4). The library transformation was recovered in 0.4 mL LB. 50 µL of transformation mixture were used to inoculate 6 mL of LB-Amp in a 15 mL plastic culture tube. This culture was allowed to shake overnight at 37 °C. The following morning, this library preculture was miniprepped using a QIAprep Spin Miniprep Kit (Qiagen, Hilden, Germany). This miniprep sample was used as the new template for mutagenesis with the primers for SSM of site 89. The Gibson products for the new five-site library were transformed using the recommended protocol into T7 Express Competent *E. coli*. Upon heat-shock transformation, mixtures were recovered in 0.4 mL Luria-Bertani medium (LB) (Research Products Int.), after which the cells were incubated at 37 °C with shaking at 220 rpm for 30 min before being plated on LB-agar plates with 100 µg/mL ampicillin (LB-amp agar plates). Single colonies from LB-agar plates were picked using sterilized toothpicks, which were used to individually inoculate 400 µL of LB containing 100 µg/mL of ampicillin (LB-amp) in 2 mL 96-well deep-well plates across 4 separate plates. The plates were incubated at 37 °C and shaken at 220 rpm for 16-18 h. The following morning 50 µL of preculture from each well were added to the wells of a 96-well flat-bottom tissue culture plate (ThermoFisher) preloaded with 50 µL of 50% glycerol solution. These glycerol stocks were stored at −80 °C for future inoculation. Additionally, the sequences of protoglobin genes contained in every well were sequenced using LevSeq sequencing[43].

### Cloning of ParPgb predicted sequences

**96-well plate gibson protocol.** Exact genes encoding ParLQ mutants predicted by Active Learning-Assisted Directed Evolution (ALDE) were synthesized and delivered by Elegen Corp. (San Carlos, CA). DNA fragments were received as dry residues in 96-well PCR plates in 2-4 µg quantities. These DNA samples were dissolved in 100 µL of double-distilled $H_2O$ (ddH$_2$O), yielding concentrations between 20-40 ng/µL. 0.7 µL of these gene solutions were added to the wells of a 96-well PCR plate (Globe Scientific Inc., Mahwah, NJ). 1.0 µL of an aqueous solution containing 60 ng/µL of linearized pET−22b(+) backbone with over-hangs designed for Gibson ligation with the ordered DNA sequences was added to each of the wells of this plate. Finally, to each well was added 5 µL of Gibson assembly mix. The 96-well plate was then incubated at 50 °C for 60 min, after which the Gibson products were placed on ice. These Gibson products could then either be directly used for transformation or stored at −20 °C for later use.

**96-well plate transformation protocol.** To each well of the previously described Gibson assembly plate was added 5 µL of T7 Express Competent *E. coli*. The cell solutions were allowed to incubate on ice for 20 min, after which they were heat-shocked at 42 °C for 10 s in a water bath. The cells were then recovered with the addition of 100 µL of LB.

Without outgrowth at 37 °C, 10 µL of each transformation mixture was used to inoculate the wells of a 2 mL 96-well deep-well plate in which the wells had been preloaded with 400 µL LB-Amp. This plate was incubated at 37 °C and shaken at 220 rpm for 16-18 h. The following morning the plate was removed from shaking and allowed to sit at room temperature for 8-10 h. After this rest phase, 1 µL from each well was used to reinoculate yet another 96-well deep-well plate preloaded with 400 µL LB-Amp. This cell passage plate was incubated at 37 °C and shaken at 220 rpm for 16-18 h. The following morning 50 µL of pre-culture from each well was added to the wells of a 96-well flat-bottom tissue culture plate (ThermoFisher) preloaded with 50 µL of 50% gly-cerol solution. These glycerol stocks were stored at −80 °C for future inoculation. The sequences of transformants generated in this manner were confirmed by LevSeq long-read sequencing.

### Protocols for the screening of ParPgb variants

**96-well plate library expression.** The wells of 2 mL, 96-well deep-well plates were filled with 400 µL LB-Amp. Previously generated 96-well plate glycerol stocks were removed from −80 °C storage and placed on dry ice. Multichannel pipet tips were used to scratch the frozen gly-cerol stock surface and used to inoculate the aforementioned deep-well plate. These pre-expression cultures were incubated at 37 °C and shaken at 220 rpm for 16-18 h. For expression cultures, the following morning 50 µL of these precultures were used to inoculate 900 µL of Terrific Broth (TB) (Research Products Int.) with 100 µg/mL of ampi-cillin (TB-amp) per well in 96-well deep-well plates. These expression cultures were initially incubated at 37 °C and 220 rpm for 2.5 h, at which point they were allowed to sit at room temperature for 30 min. Expression of proteins was induced with isopropyl-β-D-thiogalactoside (IPTG) and cellular heme production was increased with 5-aminolevulinic acid (ALA). An induction mixture containing IPTG and ALA in TB-amp (50 µL) was added to each well such that the final concentrations of IPTG and ALA were 0.5 mM and 1.0 mM, respectively. The total culture volumes were 1 mL. The plates were then incubated at 22 °C and 220 rpm overnight.

**96-well plate library reactions and screening.** Expression cultures containing *E. coli* expressing hemoproteins of interest were centrifuged at 4000 × *g* for 10 min at 4 °C. The supernatant was discarded, and nitrogen-free M9 minimal medium (M9-N, 380 µL) was added to each well. The pellets were resuspended in this medium via shaking at room temperature for 30 min. The plates were then transferred into a vinyl Coy anaerobic chamber (0–30 ppm O$_2$). To each well was added 20 µL of a MeCN solution with 200 mM of the desired styrene sub-strate and 300 mM of ethyl diazoacetate (EDA). The final reaction volume was 400 µL, and the final concentrations of the styrene and EDA were 10 mM and 15 mM, respectively. The plates were then sealed carefully with a foil cover and shaken at room temperature for 16 h in the Coy chamber. Once complete, plates were worked up for proces-sing by adding 600 µL of a 1:1 solution of ethyl acetate:cyclohexane containing 1,3,5-trimethoxybenzene as an internal standard (1.0 mM concentration). A silicone sealing mat (AWSM1003S, ArcticWhite) was used to cover the plate and the two layers were thoroughly mixed by rapid inversion of the plate. The plate was then centrifuged (5000 × *g* for 5 min at room temperature) to separate the phases. Afterwards, a 200 µL aliquot of the organic layer was transferred to a GC vial insert in a GC vial, and the samples were analyzed by GC-FID.

### Machine learning details

The initial training data for the ParPgb campaign was obtained by merging sequencing data with screening yield data. Measured yields were averaged for sequences with the same amino acid combination and normalized to the yield of the *cis* product formation of the parent variants (WYLQF) on each plate. These normalized values were used for model training and acquiring new points, which followed the same

protocol as the computational simulations on GB1 and TrpB. For the wet-lab campaign, we trained the model with onehot encodings, the DNN ensemble with 5 models and bootstrapping using 90% of the available training data for each model, and Thompson sampling as the acquisition function. These design choices correspond to the most consistent strategy based on the computational simulations. Detailed instructions on how to reproduce our results and run ALDE for other engineering campaigns are provided at https://github.com/jsunny-y/ALDE.

Most Bayesian optimization algorithms consist of two main components: (1) a probabilistic surrogate model of the objective function and (2) an acquisition function. The surrogate model predicts the objective function values at unobserved inputs, while the acquisition function quantifies the potential benefit of evaluating any given batch of inputs based on these predictions. In each iteration of the Bayesian optimization loop, a new batch of inputs is selected by maximizing the acquisition function. After evaluating the objective function at these new inputs, the surrogate model is updated, and the process repeats. Below, we describe in detail the probabilistic models and acquisition functions explored in this work, which were implemented using BoTorch[64] and GPyTorch[65].

## Probabilistic models for bayesian optimization

Let $X$ denote the input space (i.e., the space of feasible protein sequences) and let $f : X \to R$ denote the objective function (i.e., the metric we wish to optimize). In this work, we explore four classes of probabilistic surrogate models of the objective function: regular Gaussian processes (GP), deep kernel Gaussian processes (DKL), deep ensembles (DNN ensemble), and boosting ensembles.

**Gaussian Processes.** A Gaussian process model is defined in terms of a prior mean function $\mu_0 : X \to R$ and a prior covariance function $K_0 : X \times X \to R$ and it encodes a Bayesian prior distribution over $f$. Given a dataset of $n$ evaluations of the objective function, denoted as $\mathcal{D}_n = \{(x_i, y_i)\}_{i=1}^{n}$, one can derive the posterior distribution of $f$ given $\mathcal{D}_n$. If these evaluations are corrupted by i.i.d. additive Gaussian noise, i.e., $y_i = f(x_i) + \epsilon_i$, where $\epsilon_1, \ldots, \epsilon_n$ are i.i.d. Gaussian with mean zero and variance $\sigma^2$, the posterior is again a Gaussian process characterized by a posterior mean function $\mu_n : X \to R$ and a posterior covariance function $K_n : X \times X \to R$. These functions can be computed in closed form in terms of the prior mean and covariance functions as well as the data using the classical Gaussian process regression formulas[66]. The noise variance $\sigma^2$ and other hyperparameters of the model (such as the length-scale parameters) can be estimated by maximizing the log marginal likelihood.

**Deep kernel learning.** Gaussian process models with classical covariance functions, such as the Matern or squared exponential covariance functions, are known to perform poorly in high-dimensional input spaces[32]. To address this limitation, Wilson et al. (2015) proposed *deep kernel learning*[29]. Succinctly, this approach uses a covariance function of the form $K(x, x') = k(\phi_w(x), \phi_w(x'))$, where $k$ is a regular covariance function (e.g., squared exponential) and $\phi_w$ is a deep neural network with weights $w$. These weights are treated like hyperparameters of the model, which can also be estimated by maximizing the log marginal likelihood.

**Boosting ensembles.** Boosting models leverage a sequential training strategy where each new model is trained to correct the errors of the previously combined models[67]. The final prediction is often a weighted sum of the predictions made by earlier models, where the weights reflect each model's accuracy. Unlike methods such as bagging, which train models independently and in parallel, boosting specifically designs each new model to address the weaknesses of the existing ensemble, thereby creating a strong predictive model from a sequence

of weaker ones. While boosting does not inherently offer a probabilistic interpretation like Bayesian methods, it is highly effective for reducing bias and variance in predictive modeling tasks. Here, we train the boosting ensembles with bootstrapping; each ensemble consists of 5 models where 90% of the total training data is randomly seen during training.

**Deep ensembles.** Deep neural network (DNN) ensemble models are constructed by training identical deep neural network architectures multiple times, each with different random initializations of the weight parameters. Here, we train the deep ensembles with bootstrapping; each ensemble consists of 5 models where 90% of the total training data is randomly seen during training. These independently trained networks are then collectively used as if they were samples from a Bayesian posterior distribution over the objective function $f$. Unlike Gaussian processes, deep ensembles lack a proper Bayesian interpretation. However, Izmailov and Wilson argue it is possible to see these models as a form of approximate Bayesian inference[58]. We adopt this view in our work.

## Acquisition functions for bayesian optimization

**Expected improvement.** The expected improvement (EI) acquisition function is given by $\alpha_n(x) = E_n[\{f(x) - f_n^*\}^+]$, where $f_n^* = \max_{i=1,\ldots,n} f(x_i)$ and the expectation is computed with respect to the posterior distribution given $\mathcal{D}_n$[50]. For Gaussian posterior distributions and noise-free observations (where $f_n^*$ is a constant rather than a random variable), the EI can be expressed in a closed form using the posterior mean and variance. In scenarios where these conditions do not hold, computing the EI often requires approximate calculation, typically through Monte Carlo sampling techniques. When extending the EI to the batch setting, the acquisition function becomes $\alpha_n(X) = E_n[\{\max_{x \in X} f(x) - f_n^*\}^+]$, where $X = (x_1, \ldots, x_q) \in X^q$ is a batch of $q$ inputs (qEI). Maximizing the batched EI poses significant computational challenges due to the requirement to optimize over $X^q$. However, by exploiting the submodularity of the acquisition function, an efficient approximation can be achieved through a greedy optimization strategy, selecting each input in the batch sequentially. In this study, we tested qEI, but it ran slowly without noticeable improvement, so it was not included in the final results.

**Upper confidence bound.** The upper confidence bound (UCB) acquisition function is defined by $\alpha_n(x) = \mu_n(x) + \beta_n^{1/2} \sigma_n(x)$, where $\mu_n(x)$ and $\sigma_n(x)$ are the posterior mean and standard deviation, respectively, and $\beta_n$ is a parameter that controls the exploration-exploitation trade-off. In our experiments, we set $\beta_n = 4$. While there are sophisticated batch extensions of the UCB acquisition function available in the literature[68], our approach utilizes a straightforward heuristic. Specifically, we form batches by selecting the $q$ inputs that yield the highest values of $\alpha_n(x)$, evaluated across all discrete $x$ in the design space. The Greedy acquisition function can be thought of as a specific case of UCB with $\beta_n = 0$ so the acquisition function becomes $\alpha_n(x) = \mu_n(x)$. For the frequentist ensemble models, we evaluate $\mu_n(x)$ and $\sigma_n(x)$ as the mean and standard deviation of all models in the ensemble, respectively.

**Thompson sampling.** Thompson Sampling (TS) is a randomized selection strategy where the next input to evaluate is obtained by drawing a sample (function) from the posterior distribution of $f$ and selecting the point that maximizes this sample. For the GP and DKL models, we approximate samples from the posterior using 1000 random Fourier features[69]. For the frequentist ensemble models, the random function sample is drawn as one of the models in the ensemble. In the batch setting, each input in the batch is obtained as an independent sample. Unlike the EI and UCB, TS is inherently stochastic

as opposed to deterministic; however, we note that since our ensembles have five models each, TS is less stochastic in this setting.

## Reporting summary

Further information on research design is available in the Nature Portfolio Reporting Summary linked to this article.

## Data availability

All experimental and simulation data that support the findings of this study are available at https://github.com/jsunn-y/ALDE and https://zenodo.org/records/12196802. Source data are provided with this paper.

## Code availability

All code that accompanies this study is available at https://github.com/jsunn-y/ALDE under the MIT license[70].

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

## Acknowledgements

This work was supported by the U.S. Army Research Office cooperative agreement for the Institute for Collaborative Biotechnologies (W911NF-19-2-0026 to F.H.A.). J.Y. and R.G.L. are partially supported by National Science Foundation Graduate Research Fellowships. The authors thank Yueming Long and Emre Guersoy for help with sequencing and Shilong Gao for collecting useful initial data. The authors also thank Christopher Yeh for helpful discussions and Sabine Brinkmann-Chen for critical reading of the manuscript. Finally, the authors thank Miguel González-Duque and Richard Michael for pointing out consideration of the length-scale prior for GP models and Hunter Nisonoff for insight into the poor calibration of DKL models.

## Author contributions

J.Y.: conceptualization, methodology, software, investigation, analysis, writing—original draft, writing—editing. R.G.L: conceptualization, methodology, investigation, analysis, writing—original draft, writing—editing. J.C.B: methodology, software, writing—editing. R.A.: methodology, software, writing—editing. M.A.H.: investigation, writing—editing. S.K.: resources, DNA synthesis. M.H.: resources, DNA synthesis. Y.Y.: resources, writing—editing, supervision, funding. F.H.A: resources, writing—editing, supervision, funding.

## Competing interests

The authors declare no competing interests.
