## [Transparent Peer Review file · Nature Communications]

Active Learning-Assisted Directed Evolution

Corresponding Author: Professor Frances Arnold

Version 0:

Reviewer comments:

Reviewer #1

(Remarks to the Author)

In the study, the authors developed an active learning-assisted directed evolution (ALDE) workflow. They showed that the ALDE is more effective than traditional directed evolution. They also used ALDE for exploring the landscape of five epistatic residues. Despite the great effect showed in the article, several problems needed to be solved before being accepted for publication.

1. The ALDE is not a new concept. Similar studies have been reported, such as Briefings in Bioinformatics, 2023, 24(1), 1–9. A discussion about the difference between this study and other studies should be given. What is advantage of this study?
2. In the practical implementation of ALDE on ParPgb, the model used was the one determined based on computational simulations on public dataset. I was wondering if this model and encoding are generally useful for all enzymes, as the fitness landscape of different enzymes should be different? Also, logically, the computational simulations section might be better to be moved before the section of ParPgb engineering.
3. In the engineering of ParPag, the authors applied random mutations first to obtain the initial training dataset. Since the purpose of the study is to modify the enzyme selectivity and the residues targeted were selected based on previous experiments, the data obtained was fair enough to be used for training a ML model. However, for engineering an enzyme without this prior information for accepting a new substrate, a challenging task, selection of four 96-well plates random mutations might generate few positive data, which would be not enough for training a ML model. Can authors comment on this situation?
4. In the initial training dataset, does it include the single-site saturation mutagenesis performed previously? If these data were included, will the model perform better?
5. Most of the mutants in round 2 showed similar fitness with the round 1 variants, although the sequence diversity was different. What are the predicted values of these variants? Were they predicted to be better than the best variant in round 1? Can the authors provide the predicted values or ranking of the variants from different rounds?
6. What is the main difference between MLDE and ALDE? The authors have developed MLDE previously. If using MLDE, can they obtain the best variant in round 1? The users can manually add the first-round variants back to training dataset, and carry out MLDE again. So, please clarify the advantages of ALDE compared to MLDE. Maybe, give more data to support this.
7. In Figure 2C and Extended Data Figure 1, the best mutations at some sites are unclear to read. Could the authors highlight the best mutant selected at each mutation site using different colors? Additionally, I noticed several sets of missing data in Figure 2C and Extended Data Figure 1, because some amino acids were not observed in cloning and thus were not screened. Could this lead to potentially missing better mutations, and could it be considered less rigorous? Additionally, in Figures S32-S46, I noticed that the three sets of experimental data for many mutation sites vary significantly. Is such variability considered normal?
8. Lines 134-135: The authors present three variants (Figure 2D), selected based on the cis yield (DAYFW), the objective, which is the cis-trans yield (DGMDW), and the cis/trans selectivity (DHMVW). However, I cannot find the selectivity (DHMVW) in the 5th row for Cis/Trans Selectivity in Extended Data Figure 1. It appears that the best mutant at site Y57

should be G, suggesting the variant for Cis/Trans selectivity should be DGMVW. Could you clarify how DHMVW was derived, as stated in Line 135?

9. In Figure 3C, as well as Figures S48 and S49, the scatter plots for round 2 appear to be less dense compared to round 1 when observed visually. Additionally, the number of data points with a cis-Product Yield over 60% seems to have decreased. Could this indicate that the effectiveness in round 2 is worse than in round 1 and the model might not be accurate enough? While four rounds of ALDE were conducted on the GB1 and TrpB datasets, only two rounds were performed for ParPgb. What criteria were used to determine that only two rounds were necessary for ParPgb? Could you provide the scatter plot data for each round of ALDE conducted on the GB1 and TrpB datasets?

10. The authors conducted 4 rounds of ALDE on both the GB1 and TrpB datasets. However, the differences in outcomes between performing MLDE and 4 rounds of ALDE on these datasets do not seem to be highlighted. Could the authors consider adding the MLDE data for GB1 and TrpB in the Supplementary Information?

11. When applying ALDE to ParPgb, the authors use onehot encoding, DNN ensemble as the surrogate model, and Thompson sampling (TS) as the acquisition function. The article claims this setup performs well on the GB1 and TrpB datasets. However, based on computer simulation results in Figure 4, this setup did not achieve the highest activity compared to other configurations on both datasets, nor was it the most stable. The authors should consider supplementing with other high-performing model configurations for comparison, such as using AAindex encoding, boosting ensemble as a surrogate model, and TS or UCB as the sampling equation.

(Remarks on code availability)

The ALDE package can be downloaded from GitHub, and the relevant Anaconda environment can be installed using the `alde.yml` file. Running the command `python generate_domain.py --name=ParPgb --nsites=5` successfully generates the files `all_combos.csv` and `onehot_x.pt`. The analysis directory contains several results files. Could the authors provide more detailed descriptions of these files in the README file? In the last line of the README file, the authors mention the path of the raw output file as `analysis/visualization.ipynb`, but the correct file name should be `analysis/trajectory_visualization.ipynb`. Additionally, could the authors consider creating an example folder and uploading a concise example `fitness.csv` file along with related files and the correctly generated outputs? This would assist users in testing the tool with the provided data.

Reviewer #2

(Remarks to the Author)

Concerns: Yang, Lal, et al. Active Learning-Assisted Directed Evolution

The authors report on their development of an active learning method to pick better mutations and thereby make directed evolution more efficient. Active learning here means that the algorithm tells which protein variants to test next to be able to improve the predictions with the additional data. During method development, two datasets on different enzymes, with experimental results for 20⁴ mutants, were used to test which encodings, models, and acquisition functions gave the best results. This model development is a significant part of the paper. Experimentally, a protoglobin was engineered at five different positions simultaneously to improve yield and its ability to give the right diastereomer of a cyclopropane. The starting template appears to have been picked because it was not yet so good. The five selected positions were picked because in earlier engineering they were found to be important for catalytic activity. In the first round, four 96 well plates of random mutants were screened. In the two follow-up rounds, each time a 96-well plate of gene variants, as requested by the algorithm, was tested. The results were variants with improved diastereoselectivity for a collection of different target products.

The article is well written and the approach well thought through. The resulting software, ALDE, is made available at GitHub which ensures other groups can use the method as well. The study appears to be highly interesting for the protein engineering community. My main comment is that some control experiments should be added as detailed below.

Major point

1. The most important wet-lab experimental results appear to be based on single time screenings of the activities of whole cells. It appears no further control experiments were carried out to confirm the improved properties of the newly developed enzyme variants. This is unfortunate, since screenings often result in false-positives. An obvious control experiment would be to evaluate the catalytic activity and diastereomeric selectivity of the purified enzymes. At the very least, the authors should retransform the plasmid of the best variants to new E. coli cells to confirm the improvements. Also, while the single site-mutants were tested in triplo (Fig S32 to S46), this is not clearly mentioned for the final variants. If it was done, the variation (which was considerable in the first round) should be reported. It is also unclear whether the exact genes as ordered for library 2 and 3 were ever sequenced. It is presented in the discussion as an advantage that it was unnecessary (in between line 211 and 212, "the high fidelity of the delivered gene products meant that no sequencing was required"). The bottom line is that the improved properties of the best variants and their identity should be confirmed through additional experiments, if these have not yet been done.

Minor points

1. A clear limitation of the method is that it appears to require hundreds of custom synthesized gene variants with multiple mutations. Especially for larger enzymes than protoglobin, and perhaps with mutations further apart in sequence, the costs would currently be prohibitive for many academic groups and companies. At the same time, it is unclear from the manuscript how many variants should ideally be screened per round. It should be clarified why one 96-well plate was screened per round and not more or less. Did this come from the simulations with existing data (like in Extended data Fig. 5?), that 96 variants per round should be just enough or what it expected to be far more than enough? The latter seems to be suggested by the claim in the introduction ("line 92 to 93, "tens to hundreds of sequences are screened in each round"). Ideally, the authors should provide simulations with different numbers of variants screened per round, like done for 96 variants per round in Extended data Fig. 5. This would provide readers with an indication of whether the technique could be interesting when less screening per round is feasible.
2. It should be clarified why ParLQ was taken as the starting variant. The text of the manuscript suggests (line 121) that the goal was to arrive at a variant with high yield and preference for the cis-product but it appears such variants were already available in abundance from the pre-screening. Compared to the other potential starting points in Fig S31, ParLQ appears to be one of the worst with its 3:1 ratio of trans-product and only 40% overall yield. An alternative starting point, in the far-right corner of Fig S31, appears to feature some 90% yield and a roughly 23:1 preference for the desired cis-product. It should be clarified why this ParLQ was taken as a starting point. It should also be clarified which point ParLQ corresponds to in Fig. S31, since there does not appear to be a Par variant in the plot which exactly matches the description of ParLQ.
3. Some statements in the discussion seem poorly justified.
 - a) In the paragraph in between line numbers 211 and 212 (the line numbers are locally absent in the manuscript version that I have available), it is stated that greedy walk directed evolution would have required around five rounds of evolution without providing evidence for this statement. It should be clarified why it would require five rounds of directed evolution.
 - b) At line 217, it is stated that ALDE should also work on design spaces with more than five residues as long as experimental data is available on variants with mutations spread across those residues. It is unclear whether there is a logical or mathematical reason that the number of variants that need to be screened increases for example linearly (so two 96 well plates per round), and not exponentially as one could pessimistically expect. A pessimistic, but perhaps true, worst-case scenario would for example be that to screen at 10 positions instead of five positions, one would need to screen $20^{10}/20^5$ more variants to be successful with ALDE. If there is no clear logical or mathematical reason why ALDE should work fine with much more residues, the statement should be reformulated or removed.
 - c) It is presented as an advantage of ALDE that it will result in variants that are optimal for different substrates (line 207-211). This appears to be an overstatement since the variants for screening for a starting point (Figure S31) already featured many variants that were as good for production of cis-2a as for production of cis-2c till cis-2h (Figures S53 till S65). Possibly, for this particular reaction it is relatively easy to get the desired results. The statement should be better motivated or changed.
4. Part of the discussion reads like an advertisement for Elegen Corp, where two of the authors of the study are working: "Furthermore, rapid and reliable access to directly synthesized DNA (Elegen Corp.) was instrumental to the speed with which evolution was performed. The ALDE workflow was significantly enhanced with (1) the delivery of exact genes in one week, which shortened time between rounds of evolution, (2) the high fidelity of the delivered gene products meant that no sequencing was required for Rounds 1 and 2 of ALDE, and (3) no over-screening was needed because the exact sequences were arrayed individually". This should be reformulated as general features of contemporary DNA synthesis, not as the unique achievements of one company.

(Remarks on code availability)

Reviewer #3

(Remarks to the Author)

The manuscript entitled "Active-Learning Directed Evolution" proposes an active learning approach to improve traditional directed evolution models in searching for high protein fitness variants. This research is important to protein engineering and the approach proposed is a novel method compared to existing state-of-the-art directed evolution learning models. There are some minor points which can improve the computational simulation results done in this work.

1) The performance of ALDE on GB1 data set is important as GB1 serves as the primary benchmark to showcase powerful results in existing state-of-the-art directed evolution learning models (Refs. 13 and 14 in the manuscript). To further strengthen performance results of ALDE on GB1 (Figure 4D), they can be compared with existing evolutionary score-enhanced models such as Cluster learning-assisted directed evolution 2.0 (CLADE 2.0) (Ref. 14). CLADE 2.0 showed state-of-the-art performance on the GB1 data set despite the optimistic or pessimistic results, highlighting its accurate and robust performance with global maximal fitness hit rate of 88%. It is quite clear from Figure 4D that some models from each of the four encodings surpassed the maximal fitness hit rate of 88% or even 90%. It would be clearer if the numerical values can be provided.

2) The current results of ALDE also showed that the max fitness achieved is not well correlated with the calibrations of uncertainties in each model. Instead, it might be better to investigate if Deep Kernel Learning with ESM2 encoding has poor or strong correlations between its evolutionary scores and the protein fitness landscape. Existing state-of-the-art models have analyzed spearman rank correlations between evolutionary scores and protein fitness. For example, ESM-1v transformer has a low and negative correlation of -0.02 for GB1 dataset but has a high correlation of 0.48 for PhoQ dataset (see Figure 2 in Ref. 14). Based on the four models and four encodings in Figure 5, another panel can be added to discuss the correlations between ALDE's evolutionary scores and protein fitness for GB1 and TrpB dataset. The authors could potentially compare their correlations with the ones reported in CLADE 2.0 for GB1 dataset. Since ESM-2 encoding is a newer version compared to ESM-1v, it would be good to discuss any improvements in its performance (if any) under ALDE as well.

3) The authors have tested the three arbitrary sampling methods in their ALDE model, i.e. greedy walk, upper confidence bound and Thompson sampling. However, it would be interesting to know how the active learning approach affects fitness heterogeneity despite not employing any cluster-based sampling strategies. It would be valuable to understand how fitness heterogeneity changes after several mutations under ALDE.

(Remarks on code availability)

There were no issues with the code provided.
Data is provided as well.

Reviewer #4

(Remarks to the Author)

(Remarks on code availability)

The code provided contains all the models used in this work as well as instructions (README files) available to run the code for reproducibility.

Reviewer #5

(Remarks to the Author)

(Remarks on code availability)

Version 1:

Reviewer comments:

Reviewer #1

(Remarks to the Author)

Revisions are appropriate and address my earlier comments.

(Remarks on code availability)

Revisions are appropriate.

Reviewer #2

(Remarks to the Author)

The authors have sufficiently addressed all of my previous concerns.

(Remarks on code availability)

Reviewer #3

(Remarks to the Author)

The authors have addressed our concerns. We recommend the publication of their manuscript.

(Remarks on code availability)

codes are available

Reviewer #4

(Remarks to the Author)

(Remarks on code availability)

There were no issues with the code provided.
Data is provided as well.

Reviewer #5

(Remarks to the Author)

(Remarks on code availability)

N/A

Reviewer #1 (Remarks to the Author):

In the study, the authors developed an active learning-assisted directed evolution (ALDE) workflow. They showed that the ALDE is more effective than traditional directed evolution. They also used ALDE for exploring the landscape of five epistatic residues. Despite the great effect showed in the article, several problems needed to be solved before being accepted for publication.

We thank the reviewer for their extensive feedback. In addition to making general improvements throughout the manuscript, we have addressed specific questions and suggestions point-by-point below. You can find our revisions to the main text in red font.

1. The ALDE is not a new concept. Similar studies have been reported, such as Briefings in Bioinformatics, 2023, 24(1), 1–9. A discussion about the difference between this study and other studies should be given. What is advantage of this study?

Thank you for this question. We have added the following sentence to the discussion to clarify the advantages of our study: “**Compared to previous computational studies on BO for protein variants,^{14,26} our study examines a more comprehensive range of encodings, models, and acquisition functions, and it introduces analysis of the role of calibrated uncertainty quantification.**”

2. In the practical implementation of ALDE on ParPgb, the model used was the one determined based on computational simulations on public dataset. I was wondering if this model and encoding are generally useful for all enzymes, as the fitness landscape of different enzymes should be different? Also, logically, the computational simulations section might be better to be moved before the section of ParPgb engineering.

This is a good question. While it is difficult to know if the performance on one dataset translates to another dataset, we generally found that trends were consistent between the GB1 and TrpB datasets. But to address this question, we do have a recent preprint analyzing MLDE and ALDE across 16 different combinatorial protein fitness landscapes (Li et al. Evaluation of Machine Learning-Assisted Directed Evolution Across Diverse Combinatorial Landscapes, *bioRxiv* 2024).

When writing, we did consider putting the computational simulations first, but we ultimately decided to put them at the end to highlight the real-world utility of ALDE and allow readers to optionally read about the computational simulations if they are interested in those details.

3. In the engineering of ParPag, the authors applied random mutations first to obtain the initial training dataset. Since the purpose of the study is to modify the enzyme selectivity and the residues targeted were selected based on previous experiments, the data obtained was fair enough to be used for training a ML model. However, for engineering an enzyme without this prior information for accepting a new substrate, a challenging task, selection of four 96-well plates random mutations might generate few positive data, which would be not enough for

training a ML model. Can authors comment on this situation?

The reviewer is correct in their assessment that it's not trivial to select the residues to be simultaneously mutated in the ALDE workflow. However, it is our belief that ALDE will be generally useful for a set of interacting residues, and initial single-site mutagenesis screening at those residues would be informative. The exact number of mutations which can be incorporated will depend on the threshold of stability of the enzyme and the sequence-function landscape for the given reaction. We have addressed this comment in the discussion section of the manuscript: “**Determining the residues to target is an open challenge: these residues should tolerate mutations and have the potential to increase the fitness of interest. Initial domain knowledge, evolutionary conservation, or initial mutational screening may be useful here.**”

While engineering with fewer positive data is more difficult, we found in a separate preprint that ALDE is still useful on these challenging landscapes (Li et al. Evaluation of Machine Learning-Assisted Directed Evolution Across Diverse Combinatorial Landscapes, *bioRxiv* 2024).

4. In the initial training dataset, does it include the single-site saturation mutagenesis performed previously? If these data were included, will the model perform better?

The initial training dataset did not include the single-site saturation mutagenesis (SSM) results. We expect that including more data will improve model performance, but this SSM data will not always be collected. Ultimately, we did not verify this experimentally or through simulations since there are multiple backgrounds of SSM mutations that could be measured (i.e. not just the SSM mutations from parent background).

5. Most of the mutants in round 2 showed similar fitness with the round 1 variants, although the sequence diversity was different. What are the predicted values of these variants? Were they predicted to be better than the best variant in round 1? Can the authors provide the predicted values or ranking of the variants from different rounds?

This is a good question. It is correct to point out that the round 2 model may not be more accurate overall than the round 1 model, but this is not necessarily an issue for BO where the goal is only to find the maximum value. All of the queried variants are summarized in our github repo here (https://github.com/jsunny/ALDE/blob/master/analysis/ParPgb_fitness_all.csv). The exact ranking of variants can be reproduced using our code. We did not perform an extensive analysis of the predicted ranking, as we used Thompson Sampling, which means that the acquisition function considers the uncertainty of the model (not just the predicted values) and there is an element of stochasticity, as a random function is sampled from the model posterior each time.

6. What is the main difference between MLDE and ALDE? The authors have developed MLDE previously. If using MLDE, can they obtain the best variant in round 1? The users can manually

add the first-round variants back to training dataset, and carry out MLDE again. So, please clarify the advantages of ALDE compared to MLDE. Maybe, give more data to support this.

Thank you for this question. The reviewer is correct to point out that MLDE and ALDE are very similar conceptually and practically. Whereas MLDE always uses “Greedy” acquisition, the main novelty explored in ALDE is exploring whether uncertainty quantification is useful by using “UCB” or “Thompson sampling” instead. Indeed, the role of uncertainty quantification was more limited than expected (Figure 4), but ALDE is still a valuable contribution toward understanding whether batch Bayesian optimization works, and the best principles for using it. We have added the following sentence to the discussion to clarify this: “While ALDE and MLDE^{12,13} are similar conceptually and practically, MLDE only uses greedy acquisition, whereas ALDE can consider model uncertainty, which is potentially useful for exploring larger design spaces.”

7. In Figure 2C and Extended Data Figure 1, the best mutations at some sites are unclear to read. Could the authors highlight the best mutant selected at each mutation site using different colors? Additionally, I noticed several sets of missing data in Figure 2C and Extended Data Figure 1, because some amino acids were not observed in cloning and thus were not screened. Could this lead to potentially missing better mutations, and could it be considered less rigorous? Additionally, in Figures S32-S46, I noticed that the three sets of experimental data for many mutation sites vary significantly. Is such variability considered normal?

Reviewer #1 recognizes that when we screened variants that were generated through site-saturation mutagenesis, some of the single mutations were missing. To address this comment, we decided to redo the single-site mutation experiment so that none of the single mutations are missing. We have updated Figure 2C and the SI accordingly, and our findings largely remain the same. As to the high variability seen for some variants in the single site data, these reactions were run in plate format where such variability is expected due to lack of normalization of cellular OD across individual wells and considerations in reagent addition/reaction workup in plate format. The new Figure 2C is reproduced below:

8. Lines 134-135: The authors present three variants (Figure 2D), selected based on the cis yield (DAYFW), the objective, which is the cis-trans yield (DGMDW), and the cis/trans selectivity (DHMVW). However, I cannot find the selectivity (DHMVW) in the 5th row for Cis/Trans Selectivity in Extended Data Figure 1. It appears that the best mutant at site Y57 should be G, suggesting the variant for Cis/Trans selectivity should be DGMVW. Could you clarify how DHMVW was derived, as stated in Line 135?

Thank you for pointing out this misassignment. We have cloned and measured the cyclopropanation activity of the variant DGMVW. While this variant does in fact have the highest cis:trans selectivity of all naïve recombinants (~8:1), its overall

cyclopropanation yield remains low. We have replaced data for DHMVW with data for DGMVW in Figure 2D and the SI, shown below:

9. In Figure 3C, as well as Figures S48 and S49, the scatter plots for round 2 appear to be less dense compared to round 1 when observed visually. Additionally, the number of data points with a cis-Product Yield over 60% seems to have decreased. Could this indicate that the effectiveness in round 2 is worse than in round 1 and the model might not be accurate enough? While four rounds of ALDE were conducted on the GB1 and TrpB datasets, only two rounds were performed for ParPgb. What criteria were used to determine that only two rounds were necessary for ParPgb? Could you provide the scatter plot data for each round of ALDE conducted on the GB1 and TrpB datasets?

This is a good question. As mentioned earlier, it is correct to point out that the round 2 model may not be more accurate than the round 1 model, but this is not necessarily a concern in BO where the goal is not to learn the overall fitness landscape but rather to find the maximum value. Previous studies from our lab have suggested that supervised models in the low-N setting are not very accurate as measured by correlation to true values, but they are still useful for proposing more optimal variants (Wu et al. *PNAS* 2019). Because our acquisition function considers uncertainty for *exploration* (not just highest predicted value for *exploitation*), a new round of variants should not necessarily have higher fitness on average. We would note that Round 2 still proposes a few variants with fitness higher than any from the previous rounds.

We decided to only perform ALDE for two rounds, as the yield and selectivity of our optimal variant was sufficiently high (further improvement would be minimal). On more challenging design spaces, we expect that additional rounds would be needed. For GB1 and TrpB, the mean fitness achieved across 70 independent simulations is provided in Figure 4. We did not plot individual runs, as they can vary quite heavily: there are some campaigns where fitness does not improve much in a round and there are others where fitness improves significantly in a round. In short, our supervised models are not necessarily very accurate when measured by overall correlation to true values (corroborating previous studies in low-N settings), but they are still useful for optimization.

10. The authors conducted 4 rounds of ALDE on both the GB1 and TrpB datasets. However, the differences in outcomes between performing MLDE and 4 rounds of ALDE on these datasets do not seem to be highlighted. Could the authors consider adding the MLDE data for GB1 and TrpB in the Supplementary Information?

A single round of MLDE can be compared to ALDE using Figure S74.

11. When applying ALDE to ParPgb, the authors use onehot encoding, DNN ensemble as the surrogate model, and Thompson sampling (TS) as the acquisition function. The article claims this setup performs well on the GB1 and TrpB datasets. However, based on computer simulation results in Figure 4, this setup did not achieve the highest activity compared to other configurations on both datasets, nor was it the most stable. The authors should consider supplementing with other high-performing model configurations for comparison, such as using AAindex encoding, boosting ensemble as a surrogate model, and TS or UCB as the sampling equation.

We thank the reviewer for this suggestion. While we considered ensembling different models and acquisition functions, ultimately, we chose the most consistent model for simplicity. We chose the DNN ensemble because its performance fluctuated the least among different encodings. We chose the onehot encoding also because it is the most simple encoding, thus reducing the potential for artifacts resulting from feature selection. We did not use the AAindex encoding because its performance drops for the deep learning models, and we did not use the boosting ensemble as its performance drops from the higher dimensional encodings. However, we do acknowledge that different combinations of encoding and model could work the best for different systems.

Reviewer #1 (Remarks on code availability):

The ALDE package can be downloaded from GitHub, and the relevant Anaconda environment can be installed using the `alde.yml` file. Running the command `python generate_domain.py --name=ParPgb --nsites=5` successfully generates the files `all_combos.csv` and `onehot_x.pt`. The analysis directory contains several results files. Could the authors provide more detailed descriptions of these files in the README file? In the last line of the README file, the authors mention the path of the raw output file as `analysis/visualization.ipynb`, but the correct file name should be `analysis/trajectory_visualization.ipynb`. Additionally, could the authors consider creating an example folder and uploading a concise example `fitness.csv` file along with related files and the correctly generated outputs? This would assist users in testing the tool with the provided data.

These are helpful suggestions, and we have updated the github accordingly.

Reviewer #2 (Remarks to the Author):

Concerns: Yang, Lal, et al. Active Learning-Assisted Directed Evolution

The authors report on their development of an active learning method to pick better mutations and thereby make directed evolution more efficient. Active learning here means that the algorithm tells which protein variants to test next to be able to improve the predictions with the additional data. During method development, two datasets on different enzymes, with experimental results for 20^4 mutants, were used to test which encodings, models, and acquisition functions gave the best results. This model development is a significant part of the paper. Experimentally, a protoglobin was engineered at five different positions simultaneously to improve yield and its ability to give the right diastereomer of a cyclopropane. The starting template appears to have been picked because it was not yet so good. The five selected positions were picked because in earlier engineering they were found to be important for catalytic activity. In the first round, four 96 well plates of random mutants were screened. In the two follow-up rounds, each time a 96-well plate of gene variants, as requested by the algorithm, was tested. The results were variants with improved diastereoselectivity for a collection of different target products.

The article is well written and the approach well thought through. The resulting software, ALDE, is made available at GitHub which ensures other groups can use the method as well. The study appears to be highly interesting for the protein engineering community. My main comment is that some control experiments should be added as detailed below.

We thank the reviewer for their feedback and support of our study. In addition to making general improvements throughout the manuscript, we address specific questions and suggestions point-by-point below. You can find our revisions to the main text in red font.

Major point

1. The most important wet-lab experimental results appear to be based on single time screenings of the activities of whole cells. It appears no further control experiments were carried out to confirm the improved properties of the newly developed enzyme variants. This is unfortunate, since screenings often result in false-positives. An obvious control experiment would be to evaluate the catalytic activity and diastereomeric selectivity of the purified enzymes. At the very least, the authors should retransform the plasmid of the best variants to new E. coli cells to confirm the improvements. Also, while the single site-mutants were tested in triplo (Fig S32 to S46), this is not clearly mentioned for the final variants. If it was done, the variation (which was considerable in the first round) should be reported. It is also unclear whether the exact genes as ordered for library 2 and 3 were ever sequenced. It is presented in the discussion as an advantage that it was unnecessary (in between line 211 and 212, "the high fidelity of the delivered gene products meant that no sequencing was required". The bottom line is that the improved properties of the best variants and their identity should be confirmed through additional experiments, if these have not yet been done.

Thank you for these suggestions. We have sequenced all clones harboring predicted sequences using long-read sequencing (LevSeq), and all sequences were as expected. Finally, we have tested the best variants found in both rounds of ALDE in a more controlled vial format (“Small-Scale Biocatalytic or Control Reactions” protocol in SI). The function data for these variants in vial reflect the same trends in yields and selectivities seen in plate screening, with some expected variation resulting from differences in whole-cell reaction OD across the two experiments. These validated data are now included in Figure S30b (also reproduced below). We have updated the manuscript accordingly: “For all the predicted variants in Round 1 and 2 of ALDE, sequences were confirmed with LevSeq, and the yield and selectivity of the top variant from each round was validated in vial format, showing good overall consistency (Fig. S30b).” We have also clarified in the manuscript that all function tests for predicted variants were run in triplicate. Variances across replicates are now reported in the SI.

Minor points

1. A clear limitation of the method is that it appears to require hundreds of custom synthesized gene variants with multiple mutations. Especially for larger enzymes than protoglobin, and perhaps with mutations further apart in sequence, the costs would currently be prohibitive for many academic groups and companies. At the same time, it is unclear from the manuscript how many variants should ideally be screened per round. It should be clarified why one 96-well plate was screened per round and not more or less. Did this come from the simulations with existing data (like in Extended data Fig. 5?), that 96 variants per round should be just enough or what it expected to be far more than enough? The latter seems to be suggested by the claim in the introduction ("line 92 to 93, "tens to hundreds of sequences are screened in each round"). Ideally, the authors should provide simulations with different numbers of variants screened per round, like done for 96 variants per round in Extended data Fig. 5. This would provide readers with an indication of whether the technique could be interesting when less screening per round is

feasible.

Thank you for these comments. It is correct to note that gene synthesis cost is still prohibitive, but methods using oligo pools and tools like DeCOIL (Yang et al. *ACS Synth. Bio.* 2023) are viable alternatives. We believe that ALDE will be increasingly useful as the cost of gene synthesis goes down over time.

Our goal with the simulations was to compare the performance of ALDE to existing methods (such as MLDE) and to understand the impact of different encodings, models, and acquisition functions. Since the available screening budget is typically defined by assay throughput rather than as a design choice (more is always better), we decided not to include an analysis of different ALDE campaigns with different budgets. However, please note that we have a separate recent preprint analyzing ALDE, MLDE, and ftMLDE on 16 different protein fitness landscapes with different budgets, which should answer this question (Li et al. Evaluation of Machine Learning-Assisted Directed Evolution Across Diverse Combinatorial Landscapes, *bioRxiv* 2024).

2. It should be clarified why ParLQ was taken as the starting variant. The text of the manuscript suggests (line 121) that the goal was to arrive at a variant with high yield and preference for the *cis*-product but it appears such variants were already available in abundance from the pre-screening. Compared to the other potential starting points in Fig S31, ParLQ appears to be one of the worst with its 3:1 ratio of *trans*-product and only 40% overall yield. An alternative starting point, in the far-right corner of Fig S31, appears to feature some 90% yield and a roughly 23:1 preference for the desired *cis*-product. It should be clarified why this ParLQ was taken as a starting point. It should also be clarified which point ParLQ corresponds to in Fig. S31, since there does not appear to be a Par variant in the plot which exactly matches the description of ParLQ.

We selected ParLQ because it was not the highest performing protoglobin variant for any given task. It displayed relatively low cyclopropanation yields and only marginally different stereoselectivity than free heme. ParLQ was specifically selected among variants of similar activity shown in Figure S31 as it was the variant least mutated from wild-type. Furthermore, there are no known variants from *Pyrobaculum arsenaticum* (*Par*) that have high fitness for our objective, making it a challenging design task. We have updated the manuscript to clarify this: “The ParLQ variant demonstrates only moderate cyclopropanation yield (~40% yield) and stereoselectivity (3:1 preferring *trans*-**2a**) under screening conditions, and no known protein variant of *ParPgb* has high fitness for our objective.”

3. Some statements in the discussion seem poorly justified.

a) In the paragraph in between line numbers 211 and 212 (the line numbers are locally absent in the manuscript version that I have available), it is stated that greedy walk directed evolution would have required around five rounds of evolution without providing evidence for this statement. It should be clarified why it would require five rounds of directed evolution.

We have added a clarification to the discussion that typically one mutation is accumulated in each round of a DE greedy walk.

b) At line 217, it is stated that ALDE should also work on design spaces with more than five residues as long as experimental data is available on variants with mutations spread across those residues. It is unclear whether there is a logical or mathematical reason that the number of variants that need to be screened increases for example linearly (so two 96 well plates per round), and not exponentially as one could pessimistically expect. A pessimistic, but perhaps true, worst-case scenario would for example be that to screen at 10 positions instead of five positions, one would need to screen $20^{10}/20^5$ more variants to be successful with ALDE. If there is no clear logical or mathematical reason why ALDE should work fine with much more residues, the statement should be reformulated or removed.

We thank the reviewer for pointing this out. While we believe that ALDE should naturally extend to larger design spaces, it is indeed unclear how many samples/rounds will be needed, so we added the following sentence to the discussion: “Further work is needed to understand how the number of samples and/or rounds required to achieve successful engineering outcomes will increase (linearly or exponentially) with the number of sites explored simultaneously and how epistasis affects this.”

c) It is presented as an advantage of ALDE that it will result in variants that are optimal for different substrates (line 207-211). This appears to be an overstatement since the variants for screening for a starting point (Figure S31) already featured many variants that were as good for production of cis-2a as for production of cis-2c till cis-2h (Figures S53 till S65). Possibly, for this particular reaction it is relatively easy to get the desired results. The statement should be better motivated or changed.

Thanks for this comment. We have adjusted the discussion accordingly: “In the wet-lab engineering campaign, we were pleased to find that ALDE enabled access to a compilation of enzymes which, when considered together, demonstrate a broader substrate scope than that of a single enzyme. By contrast, DE is limited because it often “locks” one into high yield for only a single substrate or closely related ones, as the final variant is generally a single optimized sequence.”

4. Part of the discussion reads like an advertisement for Elegen Corp, where two of the authors of the study are working: "Furthermore, rapid and reliable access to directly synthesized DNA (Elegen Corp.) was instrumental to the speed with which evolution was performed. The ALDE workflow was significantly enhanced with (1) the delivery of exact genes in one week, which shortened time between rounds of evolution, (2) the high fidelity of the delivered gene products meant that no sequencing was required for Rounds 1 and 2 of ALDE, and (3) no over-screening was needed because the exact sequences were arrayed individually". This should be reformulated as general features of contemporary DNA synthesis, not as the unique achievements of one company.

We have corrected to the manuscript to remove mention of Elegen Corp in this sentence.

Reviewer #3 (Remarks to the Author):

The manuscript entitled “Active-Learning Directed Evolution” proposes an active learning approach to improve traditional directed evolution models in searching for high protein fitness variants. This research is important to protein engineering and the approach proposed is a novel method compared to existing state-of-the-art directed evolution learning models. There are some minor points which can improve the computational simulation results done in this work.

We thank the reviewer for their feedback and support of our study. In addition to making general improvements throughout the manuscript, we address specific questions and suggestions point-by-point below. You can find our revisions to the main text in red font.

1) The performance of ALDE on GB1 data set is important as GB1 serves as the primary benchmark to showcase powerful results in existing state-of-the-art directed evolution learning models (Refs. 13 and 14 in the manuscript). To further strengthen performance results of ALDE on GB1 (Figure 4D), they can be compared with existing evolutionary score-enhanced models such as Cluster learning-assisted directed evolution 2.0 (CLADE 2.0) (Ref. 14). CLADE 2.0 showed state-of-art performance on the GB1 data set despite the optimistic or pessimistic results, highlighting its accurate and robust performance with global maximal fitness hit rate of 88%. It is quite clear from Figure 4D that some models from each of the four encodings surpassed the maximal fitness hit rate of 88% or even 90%. It would be clearer if the numerical values can be provided.

We thank the reviewer for this suggestion. Our model performs similarly to CLADE without evolutionary scores (Figure 7A left most in CLADE 2.0), and we have added a sentence on this to the main text to clarify that our method is competitive.

Without evolutionary scores, ALDE has a lower maximal fitness hit rate than CLADE. Overall, we avoided extensive comparison to CLADE as we are most interested in understanding the role of encodings and uncertainty quantification. Still, we have provided code in the github to assist users with deeper analysis of our results.

2) The current results of ALDE also showed that the max fitness achieved is not well correlated with the calibrations of uncertainties in each model. Instead, it might be better to investigate if Deep Kernel Learning with ESM2 encoding has poor or strong correlations between its evolutionary scores and the protein fitness landscape. Existing state-of-the-art models have analyzed spearman rank correlations between evolutionary scores and protein fitness. For example, ESM-1v transformer has a low and negative correlation of -0.02 for GB1 dataset but has a high correlation of 0.48 for PhoQ dataset (see Figure 2 in Ref. 14). Based on the four models and four encodings in Figure 5, another panel can be added to discuss the correlations between ALDE's evolutionary scores and protein fitness for GB1 and TrpB dataset. The authors could potentially compare their correlations with the ones reported in CLADE 2.0 for GB1 dataset. Since ESM-2 encoding is a newer version compared to ESM-1v, it would be good to discuss any improvements in its performance (if any) under ALDE as well.

This is a good question and is very relevant to our interests in protein engineering. In general, the evolutionary scores, or likelihoods, from the ESM1b and ESM2 models

perform much better than those from ESM1v (Notin et al. *NeurIPS* 2023). Like EVmutation, we have found that these scores are positively correlated with fitness for both GB1 (Wittman et al. *Cell Systems* 2021) and TrpB (Johnston et al. *PNAS* 2024). In the past, we have also explored the utility of these zero-shot scores in ftMLDE (Wittman et al. *Cell Systems* 2021) and confirmed in CLADE for GB1.

In this study, we decided not to study zero-shot scores extensively and instead focus on the role of uncertainty quantification so that ALDE can be applied to any dataset – in fact we found that typical zero-shot scores would not likely have helped with our real-world ParPgb engineering campaign (Figure S72). That being said, we have a recent preprint analyzing the role of zero-shot scores and MLDE/ALDE across 16 protein fitness landscapes, and the suggested analysis is a key component of this study (Li et al. Evaluation of Machine Learning-Assisted Directed Evolution Across Diverse Combinatorial Landscapes, *bioRxiv* 2024). We found here that using zero-shot scores does in fact improve performance ALDE performance, and different zero-shot scores are useful for different landscapes.

3) The authors have tested the three arbitrary sampling methods in their ALDE model, i.e. greedy walk, upper confidence bound and Thompson sampling. However, it would be interesting to know how the active learning approach affects fitness heterogeneity despite not employing any cluster-based sampling strategies. It would be valuable to understand how fitness heterogeneity changes after several mutations under ALDE.

Thank you for this suggestion. We did not do an extensive study of fitness heterogeneity as the analysis does not seem to be readily provided in the CLADE code. However, it will be interesting for future extensions of CLADE to analyze fitness heterogeneity in comparison to various acquisition functions.

Reviewer #3 (Remarks on code availability):

*There were no issues with the code provided.
Data is provided as well.*

Reviewer #4 (Remarks to the Author):

Reviewer #4 (Remarks on code availability):

The code provided contains all the models used in this work as well as instructions (README files) available to run the code for reproducibility.

Reviewer #5 (Remarks to the Author):
